# Psychometric properties of the Probability Bias Measure

Robert W. Booth *, Gubse N. Aydın, Beril Başara, Ceren Yılmaz

Faculty of Arts and Social Sciences, Sabanci University, Istanbul, Turkey

* rob.booth.psych@gmail.com

## Abstract

Probability bias is the tendency to think that negative events are relatively likely to happen, and that positive events are relatively unlikely to happen. It is reliably associated with depressive and anxious symptom severity. Here, we describe the initial development and testing of the Probability Bias Measure, which is designed to be relatively brief and general, allowing its convenient use with different populations. In two samples of Turkish-speaking students ($N$s = 228 and 170), probability estimates for 12 positive and 12 negative events both displayed adequate one-factor structure, and good internal consistency. More importantly, probability bias – the difference between a participant's mean probability estimate for negative events, and their mean probability estimate for positive events – showed excellent convergent validity, in that it correlated with depressive and anxious symptom severity, positive and negative mood, hopelessness, and dispositional optimism. Furthermore, probability bias accounted for additional variance in depressive symptom severity, over and above that already accounted for by hopelessness and dispositional optimism, which supports its discriminant validity. These findings provide preliminary support for the validity of the Probability Bias Measure for research with Turkish-speaking populations; we also discuss initial evidence for the validity of its English version. The full measure is provided in Turkish and English.

## Introduction

Most people, most of the time, are irrationally optimistic [1–3]. This optimism takes many forms: one reliable finding is that people feel that positive events are more likely to happen to them than negative events [e.g., 4–6]. However, an equally reliable finding is that this optimistic bias is reduced in people with depressive or anxious symptoms; people with more severe symptoms may even feel that negative events are more likely to happen to them [7–9]. We use the term 'probability bias' (it is also known as 'expectancy bias') to refer to this tendency for people with more severe symptoms to feel that negative events are relatively probable, and positive events relatively improbable, compared to people with less severe symptoms [10,11].

**Data availability statement:** Anonymised datasets are freely available at https://doi.org/10.17605/OSF.IO/GXJK8.

**Funding:** The author(s) received no specific funding for this work.

**Competing interests:** The authors have declared that no competing interests exist.

Probability bias is a form of judgement bias [12,13], and is conceptually related to other future-thinking phenomena reported in depression, anxiety, and suicidality [see, e.g., 14–18].

### Relationships between probability bias and symptoms

Probability bias is a reliable and robust correlate of depressive symptoms and, to a lesser extent, anxious symptoms. These relationships are substantial: we routinely observe correlations of.40 or more between probability bias and depressive symptom severity, and of.30 or more with anxious symptom severity (although the relationship with anxiety seems to be indirect; [11,19,20]). Despite these appealing qualities, and despite the emphasis which Beck [e.g., 21] placed on pessimistic beliefs in depression and anxiety, probability bias has attracted less research attention than other cognitive phenomena associated with these disorders [e.g., 22–25].

Since Beck [21], most theoretical interest in probability bias has come from anxiety researchers. In particular, Foa and colleagues [e.g., 26] have long suggested that probability bias is an important component of the pathological fear structures held by people with social anxiety. Relatedly, Rapee and Heimberg [e.g., 27] have conceptualised probability bias as a proximal cause of symptoms when people with social anxiety are watched or evaluated. More generally, Grupe and Nitschke [28] assumed exaggerated probability bias contributes to the hypervigilance, emotional reactivity, and avoidance behaviours seen in anxious people.

Consistent with these models, probability bias has been reported in people with several anxiety and anxiety-related disorders [29–33]; it has also been reported in people with depressive disorders [34–38]. Importantly, there is also evidence that probability bias may indeed play a causal role in symptom development and progression: pessimistic expectations about the future appear to be longitudinal predictors of depressive symptoms, suicidal ideation and self-harming behaviour [39]. Congruent with this idea, changes in probability bias can mediate improvements in anxiety symptoms during treatment [40–44]. Furthermore, psychotherapies which specifically target future thinking can be effective [45,46]. Crucially, probability bias – and its relationships with symptoms, especially depressive symptoms – is empirically distinguishable from related constructs, such as dispositional optimism, pessimistic explanatory style, and repetitive negative thinking [10,11,47].

From this brief review, we can conclude that probability bias is a useful construct. It is useful theoretically, in that cognitive models of psychopathology view probability bias as a predictor, and perhaps a determinant, of symptoms. It is also useful practically, in that changes in probability bias can partially account for symptom improvement during treatment, and treatments which target biased future thinking show some efficacy. Therefore, it is important to identify good-quality, effective instruments for measuring probability bias.

### Measuring probability bias

A single, dominant approach to measuring probability bias has yet to emerge. Measures have tended to be developed on an ad hoc basis, for individual projects or

by individual research groups; sometimes detailed psychometric data are lacking for these measures (although see, e.g., [6,48,49]). Some measures have only assessed estimated probabilities for negative events [50], while others have assessed probabilities for both positive and negative events [51]; some have used numeric response scales, others have used purely verbal scales [52]; different measures have asked about the probability of events happening within different time periods, ranging from a few minutes later [53] to within the next year [11], have used different time periods for different events [54], or have not specified a time period at all [55,56]; and some measures have examined specific categories of events, such as social situations, or physical illnesses [32,57]. This proliferation of different measures and approaches has made it challenging to compare the results of different studies, and challenging to choose the most appropriate measure for new studies. We therefore decided to create and test our own standard measure, which would provide consistency for our own future studies. This measure could be designed from the bottom up for use with the diverse Turkish-speaking samples we use in our research. (It can also be used with English-speaking samples; see General Discussion).

## The Probability Bias Measure

We developed the Probability Bias Measure (see S1 File. Full Measure) with several criteria in mind:

Firstly, we wanted the measure to be relatively brief, so that it could be easily deployed alongside other measures, in a variety of contexts.

Secondly, the measure needed to include both positive and negative events. There is some evidence that estimated probabilities for positive and negative events can show different relationships with psychopathology [58]. Furthermore, probability bias – the difference between estimated probabilities for positive and negative events – often shows stronger relationships with symptom severity than either of the raw probabilities do [20]. This is partly attributable to the superior psychometric properties of a difference score, when it takes a difference between negatively-correlated scores (see Supplementary Materials of [19]).

Thirdly, we wanted the events to be as general as possible, so that the measure could be used with different populations. Therefore, we avoided events related to education or early career stages, and events that might have different connotations for, for example, older participants.

Fourthly, it was important to use a verbal response scale, with no numbers at all. There is evidence [52,59] that probability bias is weakened when participants are asked to make numerical judgements of probability (e.g., '75% chance of happening') compared to when they make more general verbal judgements (e.g., 'quite likely to happen'). Numbers may elicit a more analytical, mathematical approach to the task (see also [60]).

Finally, we note that we randomise the order of events for each participant, and events are presented serially, i.e., participants respond to one event before they see the next one. This is because we conceptualise our measure, which involves a judgement of probability rather than a rating of agreement, as somewhat straddling the line between self-report scale and behavioural cognitive task. We also wish to avoid the risk that responses to earlier events will confound responses to later events. Since we typically collect data using online survey platforms, presenting events serially in a random order is straightforward. Other researchers may prefer to use our measure with a consistent (mixed) order of events if, for example, they are using pen-and-paper surveys: we recommend they pay extra attention to the psychometric properties of the measure in their study, since we have not tested it in this format.

## The present studies

Here we report two studies on the psychometric properties of the Probability Bias Measure. In Study 1, we tested a large initial set of events, and selected a set of 12 positive and 12 negative events for further testing. In Study 2, we assessed the test-retest reliability and construct validity of our selected events.

## Study 1

In Study 1, we tested a large set events in order to select those with the most satisfactory characteristics. We wished to develop a brief measure of probability bias for research use; we were less interested in whether the events could be grouped into different categories. Therefore, we were more interested in the reliability and validity of our measure than we were in its factor structure. Nevertheless, we did examine the factor structure in order to be sure we had appropriately categorised events as 'positive' and 'negative'.

### Method

The studies reported here were approved by the Sabanci University Research Ethics Committee, approval numbers FASS-2022-58 and FASS-2023-22.

**Participants.** Two hundred and fifty-six Sabanci University students participated for course credit. All were native Turkish speakers. Twenty-three failed an attention check at the end of the probability questions (see below), and their data were excluded. Six (1 of which had also failed the attention check) reported a current psychiatric diagnosis other than a DSM-5 depressive or anxiety disorder (4 attention-deficit/hyperactivity disorder; 1 borderline personality disorder; 1 obsessive-compulsive disorder), and were excluded. This left 228 participants for analysis (135 women, 92 men, 1 other), with a mean age of 22 ($SD = 2$). Four reported current diagnoses of major depressive disorder or simply 'depression', six reported generalised anxiety disorder or simply 'anxiety', one reported social anxiety disorder, four reported both depression and anxiety, one reported both depression and social anxiety, and one reported both depression and panic attacks.

**Measures.** Alongside the probability estimates, participants also completed measures of depression, anxiety, and positive and negative mood.

Participants were presented with 127 events, 41 of which were positive and 86 were negative. These were adapted from published research on probability bias [e.g., 7,11,20,51,55,61–63], and covered a range of topics including health and injury, crime, social acceptance or rejection/embarrassment, professional success or failure, financial success or failure, and family and romantic relationships. Events were chosen which we felt would be relevant to a majority of people: for example, events related to university life were excluded. Events that were not already in Turkish were translated by one of the native Turkish-speaking authors, and back-translated by another. Discrepancies were resolved by discussion, and both authors approved the final list. During the translation process, these authors also ensured that events were appropriate to the Turkish culture. See S2 File. Additional Events for a full list of the events, in both English and Turkish.

Participants rated the probability of each event happening to them, using a 7-point verbal scale. The scale points were labelled: Would never happen to me; Would probably not happen to me; Might not happen to me; Might happen, might not; Might happen to me; Would probably happen to me; Would definitely happen to me. Responses were converted to a 1–7 numerical scale for analysis.

At the end of the probability events, participants were given an additional event, as an attention check. This event was "If you are still paying attention, please select 'Would definitely happen to me'." Participants giving the incorrect response were excluded from all analyses.

Depressive symptom severity was measured with the Beck Depression Inventory [64, trans. 65]. This presents participants with 21 groups of four statements, each of which assesses a symptom of depression, with the first statement indicating the absence of the symptom, and the others indicating progressively more severe presence of the symptom. These four statements are scored 0–3. Participants are asked which statement best describes their situation over the previous 2 weeks. Internal consistency was very good in this sample, McDonald's $\omega = .89$, 95% CI [.87, .91]. The mean score was 13.1 ($SD = 8.8$), within the typical range for unselected student samples.

Anxious symptom severity was measured with the Beck Anxiety Inventory [66,67]. This asks participants how much they have been 'bothered' by 21 symptoms over the previous 1 week. Participants respond on a 0–3 scale for each

symptom. Internal consistency was very good, ω = .90, 95% CI [.88,.92]. The mean score was 14.3 (*SD* = 10.3), which is within the typical range for unselected student samples.

Mood was measured with the Positive And Negative Affect Schedule (PANAS; [68,69]). Despite its name, the PANAS is designed to measure positive and negative activation, not affect. Given the diversity of its items, and the fact that it measures the participant's general state over the preceding two weeks, we feel it is best characterised as a measure of mood. The PANAS presents participants with 10 positive and 10 negative affective states, and asks how much they have experienced each one in the preceding two weeks. Participants respond on a 1–5 scale. Internal consistency was very good: for positive mood ω = .90, 95% CI [.88,.92]; for negative mood ω = .87, 95% CI [.83,.89]. The mean score was 28.5 (*SD* = 8.2) for positive mood, and 22.5 (*SD* = 7.6) for negative mood.

**Procedure.** The data were collected online, between the 2nd and 19th of January 2023. Participants were told the study was about how people think about the probabilities of various events happening to them. After clicking to approve a written consent form, they completed the probability estimates. Then, they completed the measures of depressive symptom severity, anxious symptom severity, and mood, in a random order. They were then debriefed.

## Results

Anonymised datasets, and SPSS syntax for computing scale scores, are freely available at https://doi.org/10.17605/OSF.IO/GXJK8. When calculating scale scores, including the total scores for the selected probability bias events (see below), missing responses were replaced with the participant's mean response to that scale. If a participant missed more than two responses on a single scale, their whole score for that scale was coded as missing. All analyses were conducted on all available data, i.e., listwise deletion was not used.

**Item selection.** We began by running principle axis factor analyses on all the positive events, and all the negative events. Promax rotation was applied. Since the responses were ordinal variables, these factor analyses were based on the polychoric correlation matrices.

The positive events showed the presence of a latent structure, MSA = .86, Bartlett's $\chi^2$ (820) = 4753.21, $p < .001$. Parallel analysis suggested the presence of three factors, explaining 38% of the variance; however, we were unable to see a clear interpretation for the difference between them, and there were many cross-loaded items. Furthermore, the eigenvalue for the first factor (12.38) was clearly larger than the eigenvalues for the other factors (2.84 and 2.07), and the correlations among the factors were strong (all correlations ≥ .50). We therefore concluded that there was one dominant factor, reflecting a general optimism about positive future events. To select 12 events for the finished scale, we re-ran the factor analysis, constraining it to a single factor (eigenvalue = 12.38; accounting for 29% of the variance), and selected events with good loadings, clear correlations with symptom severity and mood, and (as much as possible) diverse topics. Results of a reduced analysis of these 12 events are presented in Table 1; the items had a one-factor structure (eigenvalue = 5.37; accounting for 40% of the variance), and all items had loadings > .40 on that factor.

The negative events also showed a latent structure, MSA = .82, Bartlett's $\chi^2$ (3655) = 10,211.35, $p < .001$. Parallel analysis suggested six factors were present, accounting for 36% of the variance; although again, the first factor's eigenvalue (17.39) was substantially larger than those of later factors (which descended from 5.78 to 2.33). The first factor seemed most associated with social misfortune, the second factor with physical injury and illness, and the third factor with incompetence and losing things. However, we required the negative events to reflect a general pessimism about future negative events, so we again selected 12 events by constraining the analysis to a single factor (eigenvalue = 17.39; accounting for 19% of the variance), and selecting events with good loadings, clear correlations with symptom severity and mood, and diverse topics. Note, however, that in order to ensure we selected the best-performing events, we had to include a relatively high proportion of socially-relevant events. Results of a reduced analysis on these 12 events are presented in Table 1; the selected items had a one-factor structure (eigenvalue = 4.75; accounting for 34% of the variance), and all items had loadings > .35 on that factor.

**Table 1. Factor loadings for selected items, Study 1. Positive and negative events were analysed separately.**

| Positive Events | | | Negative Events | | |
|---|---|---|---|---|---|
| Item number | Loading | Uniqueness | Item number | Loading | Uniqueness |
| P1 | .70 | .51 | N21 | .48 | .77 |
| P7 | .70 | .50 | N27 | .70 | .51 |
| P10 | .71 | .50 | N28 | .61 | .63 |
| P14 | .59 | .65 | N30 | .57 | .68 |
| P17 | .64 | .59 | N31 | .50 | .75 |
| P19 | .56 | .69 | N36 | .70 | .51 |
| P23 | .57 | .68 | N47 | .41 | .83 |
| P25 | .61 | .62 | N51 | .62 | .62 |
| P26 | .73 | .46 | N58 | .64 | .59 |
| P27 | .67 | .55 | N70 | .36 | .87 |
| P29 | .40 | .84 | N80 | .68 | .53 |
| P30 | .62 | .62 | N81 | .65 | .58 |

**Reliability and validity of selected items.** The 12 positive events showed very good internal consistency, $\omega = .87$, 95% CI [.84, .89], and their sum ($M = 56.1$, $SD = 9.9$) correlated as expected with the other measures. The 12 negative events also showed good internal consistency, $\omega = .84$, 95% CI [.81, .87], and their sum ($M = 49.9$, $SD = 10.4$) correlated as expected with the other measures too. Most importantly, the probability bias index – the mean probability of negative events, minus the mean probability of positive events ($M = -6.2$, $SD = 18.2$) – showed correlations of the expected magnitude and direction with the other measures. This is important because this index score can show stronger and more consistent correlations with other variables than the raw positive or negative estimates do (e.g., [20]). Probability estimates for positive events, estimates for negative events, and the bias index's correlations with the other study variables are presented in Table 2.

**Table 2. Correlations between total probability estimates for positive and negative events, and probability bias, and the other measures.**

| Variables | | r | 95% CI | N |
|---|---|---|---|---|
| Probability of positive events | Depressive symptom severity | −.55 | −.63, −.45 | 228 |
| | Anxious symptom severity | −.26 | −.38, −.14 | 227 |
| | Positive mood | .58 | .49,.66 | 226 |
| | Negative mood | −.31 | −.42, −.19 | 226 |
| Probability of negative events | Depressive symptom severity | .64 | .55,.71 | 228 |
| | Anxious symptom severity | .52 | .42,.61 | 227 |
| | Positive mood | −.52 | −.61, −.42 | 226 |
| | Negative mood | .52 | .42,.61 | 226 |
| Probability bias | Depressive symptom severity | .66 | .58,.73 | 228 |
| | Anxious symptom severity | .44 | .33,.54 | 227 |
| | Positive mood | −.62 | −.64, −.46 | 226 |
| | Negative mood | .47 | .36,.57 | 226 |

All correlations are significant, $p < .001$. CI = confidence interval.

## Discussion

Study 1 successfully identified 12 positive and 12 negative events for inclusion in the final measure. Both sets of events showed acceptable factor structure and very good internal consistency, and correlated as expected with the other measures. However, our negative events were disproportionately focused on social misfortunes or embarrassments. Our next step was to more-thoroughly test the reliability and validity of these events in a second sample.

For both positive and negative events, a single primary factor seemed to emerge. This may imply that optimism/pessimism is a general phenomenon, which influences participants' probability estimates for different types of events to a similar extent [57]. However, this finding may have been biased by our relatively-healthy sample: for example, people with social anxiety disorder might show clearer differences between their estimates for socially-related vs. socially-unrelated events [55,70,71].

## Study 2

In Study 2, we tested the reliability and validity of the Probability Bias Measure in a fresh sample of students.

### Method

**Participants.** One hundred and seventy-six Sabanci University students participated at Time 1. Three reported a current psychiatric diagnosis other than a DSM-5 depressive or anxiety disorder (one bipolar disorder, one obsessive-compulsive disorder, and one attention-deficit/hyperactivity disorder), and three failed an attention check at the end of the probability bias measure, leaving 170 (105 women, 60 men, 5 did not respond) participants for analysis, with a mean age of 21 ($SD = 2$). Three reported a diagnosis of depression, one a diagnosis of minor depression, four reported diagnoses of anxiety, and one reported suffering with exam anxiety and stress. They were asked to indicate their ethnicity, with a free response: 145 described themselves as Turkish or partly Turkish, one as Kurdish, two as Azerbaijani, two as being from the Caucuses, three as Tatar, and one as Balkan. They were also asked to indicate their subjective socioeconomic status on a 10-rung ladder, where 1 is the lowest income, education and occupations, and 10 is the highest income, education and occupations [72]: the mean response was 7.0 ($SD = 1.3$). Not all participants completed all the Time 1 measures, so the $N$s for individual analyses below vary.

One hundred and twenty-seven participants returned for Time 2. (Forty-nine dropped out; there were no significant differences between returning participants and dropouts on any demographic or Time-1 study variables.) Two of the participants who had reported a diagnosis other than a depressive or anxiety disorder returned, and were excluded. A further four failed the attention check at the end of the probability measure, and were excluded, leaving 121 participants for analysis of Time 2 measures. One of these participants joined Time 2 too late, more than 5 weeks since they completed Time 1, and their data were also excluded from analyses using both timepoints: such analyses are therefore based on 120 participants.

**Measures.** At Time 1, participants completed the probability measure, and measures of hopelessness and dispositional optimism. They also completed the measures of depressive and anxious symptom severity, and the mood measure; these were the same as for Study 1, except that the mood measure was adjusted to measure state mood, at the time the participant completed the measure. At Time 2, participants again completed the probability measure, the measures of depressive and anxious symptom severity, and the state mood measure.

Participants were presented with the 12 positive and 12 negative events which were chosen based on the results of Study 1. They were presented in a random order for each participant. The instructions and response scale were identical to those used in Study 1. The attention check was included at the end of the probability measure.

Hopelessness was assessed with the Beck Hopelessness Scale [73, trans. 74]. This asks participants to answer either Yes or No to 20 items, including 'I might as well give up because I can't make things better for myself'. For each item, the more hopeless answer is scored 1, the less hopeless answer is scored 0. Internal consistency was very good in this sample, $\omega = .87$, 95% CI [.83,.90]. The mean score was 5.1 ($SD = 4.2$), which indicates 'mild' hopelessness.

Dispositional optimism was assessed with the Life Orientation Test – Revised [75]. This asks participants how much they agree with six items such as 'In uncertain times, I usually expect the best', on a 0–4 scale. Four additional items are included as fillers. We adapted Aydın and Tezer's [76] translation of the original Life Orientation Test, adding Karacan Özdemir's [77] translation of the additional item in the revised scale, and Kahleoğulları's response scale [78]. Internal consistency was good in this sample, $\omega = .81$, 95% CI [.77,.85]. The mean score was 12.4 ($SD = 4.5$).

The scales of depressive and anxious symptom severity, and of positive and negative mood, all showed good internal consistency as before. At Time 1, for depressive symptom severity, $\omega = .85$, 95% CI [.81,.88], $M = 12.0$, $SD = 7.7$; for anxious symptom severity, $\omega = .92$, 95% CI [.89,.94], $M = 14.9$, $SD = 11.3$; for positive mood, $\omega = .91$, 95% CI [.87,.92], $M = 27.8$, $SD = 8.6$; for negative mood, $\omega = .84$, 95% CI [.81,.87], $M = 17.5$, $SD = 6.2$. At Time 2, for depressive symptom severity, $\omega = .88$, 95% CI [.85,.91], $M = 10.1$, $SD = 7.7$; for anxious symptom severity, $\omega = .87$, 95% CI [.84,.90], $M = 12.9$, $SD = 8.7$; for positive mood, $\omega = .91$, 95% CI [.87,.93], $M = 26.0$, $SD = 8.6$; for negative mood, $\omega = .90$, 95% CI [.87,.93], $M = 18.2$, $SD = 7.4$.

**Procedure.** The data were collected online, between 10[th] April 2023 and 5[th] January 2025. Participants were told that the study was about how people think about the future. They signed up for both parts of the study at the same time. At Time 1, they clicked to accept a written consent form, and then completed the Probability Bias Measure, followed by the measures of depressive and anxious symptom severity, mood, dispositional optimism and hopelessness, in a random order.

Participants were reminded by email to complete the second part of the study four weeks after completing the first part. They were required to complete the second part within one week of this reminder. At Time 2, they completed the Probability Bias Measure, followed by the measures of depressive symptom severity, anxious symptom severity, and mood, in a random order. They were then debriefed.

## Results

**Latent structure of the Probability Bias Measure.** We conducted separate principle axis factor analyses on the positive and negative events, for both timepoints, based on the polychoric correlation matrices.

At Time 1, the positive events showed a latent structure, MSA = .87, Bartlett's $\chi^2$ (66) = 892.60, $p < .001$. Parallel analysis and the scree plot indicated there was one main factor, with an eigenvalue of 5.44, which explained 41% of the observed variance. All events had loadings > .48 on that factor (see Table 3). The negative events also showed a latent structure, MSA = .88, Bartlett's $\chi^2$ (66) = 934.39, $p < .001$. Again, parallel analysis and the scree plot indicated there was one main factor, with an eigenvalue of 5.54, which explained 42% of the observed variance. All events had loadings > .46 on that factor (see Table 3).

At Time 2, the positive events showed a latent structure, MSA = .86, Bartlett's $\chi^2$ (66) = 771.18, $p < .001$. Parallel analysis and the scree plot indicated there was one main factor, with an eigenvalue of 5.97, which explained 45% of the observed variance. All events had loadings > .53 on that factor (see Table 3). The negative events also showed a latent structure, MSA = .86, Bartlett's $\chi^2$ (66) = 745.76, $p < .001$. Again, parallel analysis and the scree plot indicated there was one main factor, with an eigenvalue of 5.70, which explained 43% of the observed variance. All events had loadings > .36 on that factor (see Table 3).

**Reliability.** Both positive and negative events showed very good internal consistency. At Time 1, for positive events $\omega = .86$, 95% CI [.83,.89], and for negative events $\omega = .88$, 95% CI [.85,.91]. At Time 2, for positive events $\omega = .88$, 95% CI [.85,.91], and for negative events $\omega = .90$, 95% CI [.87,.92].

The positive and negative events showed excellent test-retest reliability, as did the probability bias index (mean probability of negative events minus mean probability of positive events). For positive events, $r$ (118) = .83, 95% CI [.77,.88], $p < .001$; for negative events, $r$ (118) = .86, 95% CI [.81,.90], $p < .001$; for probability bias, $r$ (118) = .88, 95% CI [.84,.92], $p < .001$.

**Table 3. Factor loadings for probability estimates, Study 2. Positive and negative events were analysed separately, at both timepoints.**

**Positive Events**

| Event | Time 1 | | Time 2 | |
| --- | --- | --- | --- | --- |
| | Loading | Uniqueness | Loading | Uniqueness |
| You will become well-known for an outstanding accomplishment | .76 | .42 | .67 | .55 |
| You will be very well-known in your field | .73 | .47 | .68 | .54 |
| You will be completely satisfied with your life | .70 | .50 | .78 | .39 |
| Tomorrow will be a wonderful day for you | .66 | .56 | .66 | .57 |
| You meet some new people, and make a good impression | .64 | .59 | .70 | .51 |
| You receive compliments about your appearance | .52 | .73 | .53 | .72 |
| You will have a wonderful 90th birthday | .49 | .76 | .55 | .69 |
| You will have lots of energy and enthusiasm | .67 | .56 | .72 | .48 |
| You will achieve the things you set out to do | .74 | .46 | .75 | .44 |
| You will be very fit and healthy | .54 | .71 | .67 | .55 |
| You will be able to cope easily with pressure | .51 | .73 | .66 | .56 |
| Your mind will be very alert and 'on the ball' | .61 | .63 | .66 | .56 |

**Negative Events**

| Event | Time 1 | | Time 2 | |
| --- | --- | --- | --- | --- |
| | Loading | Uniqueness | Loading | Uniqueness |
| You will make a decision which you later regret | .53 | .72 | .59 | .65 |
| People will find you boring | .67 | .56 | .64 | .59 |
| People will think you're a failure | .72 | .49 | .72 | .48 |
| You will fall badly behind in your work | .57 | .68 | .67 | .55 |
| You will be unable to confide in anyone | .56 | .68 | .56 | .68 |
| You will feel stupid while talking to others | .74 | .45 | .83 | .31 |
| Your mind will stop functioning normally | .49 | .76 | .57 | .68 |
| You will be criticised for poor performance | .70 | .51 | .66 | .56 |
| You will feel flustered in front of others | .76 | .42 | .66 | .56 |
| You will go crazy | .46 | .79 | .36 | .87 |
| You will feel inferior to others | .74 | .46 | .73 | .46 |
| You will be unable to express yourself in social situations | .70 | .51 | .76 | .43 |

**Validity.** We assessed the convergent validity of the probability bias index at both timepoints. It showed significant correlations of the expected magnitude and direction with all the other scales (see Table 4).

We assessed discriminant validity somewhat indirectly. Booth and Sharma [47] recently reported, using similar measures in a sample of UK students, that the relationship between probability bias and depressive symptom severity could not be accounted for by dispositional optimism. In other words, probability bias accounted for additional variance in depressive symptoms, over and above that explained by dispositional optimism. We replicated and extended that finding, by checking the partial correlation between probability bias and depressive symptom severity at Time 1, while controlling for both dispositional optimism and hopelessness. A confidence interval was estimated from 1000 bootstrap replications. This partial correlation was significant and not insubstantial, $r$ (159) =.32, 95% CI [.19,.45], $p < .001$. Next, we tested the partial correlation between probability bias and positive mood at Time 1. Again, the partial correlation was significant and not insubstantial, $r$ (159) = −.31, 95% CI [−.47, −.14], $p < .001$ (neither analysis showed any evidence of problematic multicollinearity). These results show that, despite the strong correlations between probability bias and both dispositional optimism and hopelessness ($r$s = −.63 and.62 respectively), probability bias still accounts for additional variance in depressive

**Table 4. Bivariate correlations between probability bias index and the other scales, at both timepoints.**

| Time 1 | | | | |
|---|---|---|---|---|
| **Variable** | *r* | **95% CI** | *N* | *p* |
| Hopelessness | .62 | .52,.71 | 164 | <.001 |
| Dispositional optimism | −.63 | −.72, −.53 | 166 | <.001 |
| Depressive symptom severity | .59 | .48,.68 | 164 | <.001 |
| Anxious symptom severity | .34 | .19,.46 | 165 | <.001 |
| Positive mood | −.53 | −.67, −.47 | 164 | <.001 |
| Negative mood | .43 | .30,.55 | 164 | <.001 |
| **Time 2** | | | | |
| **Variable** | *r* | **95% CI** | *N* | *p* |
| Depressive symptom severity | .51 | .36,.63 | 122 | <.001 |
| Anxious symptom severity | .28 | .11,.44 | 121 | .002 |
| Positive mood | −.59 | −.69, −.46 | 121 | <.001 |
| Negative mood | .24 | .06,.40 | 121 | .008 |

*Note.* CI = confidence interval.

symptoms and positive mood, over and above that explained by these other variables. This supports the discriminant validity of the Probability Bias Measure.

## Discussion

The Probability Bias Measure showed very good psychometric properties. Probability estimates for the positive and negative events both showed good internal consistency; the probability bias index, which can show more consistent correlations with other variables than the probability estimates themselves (e.g., [20]), showed excellent test-retest reliability and convergent validity. The Measure's psychometric properties were comparable or favourable to those of other similar instruments [6,48,49].

We also replicated Booth and Sharma [47], in showing that probability bias accounts for additional variance in depressive symptom severity over and above that accounted for by hopelessness and (low) dispositional optimism. This supports the discriminant validity of our measure, and of probability bias as a construct. It suggests that different biases in affect-related future thinking – different varieties of optimistic vs. pessimistic beliefs and expectancies – are empirically distinguishable, and can have different relationships with other individual differences and symptoms (see also [79]).

Probability estimates for both positive and negative events seemed to show one-factor structures in Study 2. Their structures were both clearer and simpler than they appeared to be in Study 1, especially for the negative events. Perhaps the larger set of events in Study 1 encouraged participants to consider the differences between their categories: for example, the differences in probability between social misfortunes and physical injuries. In contrast, the reduced set of events in Study 2 may have encouraged participants to think about positive vs. negative events in more general terms. Our measure is intended to gauge general beliefs about the future, and the results of Study 2 suggest it is well-suited for this purpose, although it must be noted that many of the negative events are socially relevant. However, under certain conditions, people's probability estimates may vary between different categories of events: people with social anxiety may believe that social misfortunes are more probable than other types of negative events, for example. The evidence for such specificity of probability bias is unclear at this time [55,57,70,71].

## General discussion

These results offer preliminary support for the reliability and validity of the Probability Bias Measure, for the assessment of the biased expectations about the future which are associated with depressive symptoms. The factor structures of the positive and negative events are acceptable, but less clear than we expected: for both groups of events, one main factor emerges, but this factor accounts for less than half of the observed variance in participants' estimates. This implies that there are other variables, beyond simple optimism/pessimism, which influence probability estimates. These may include participants' temporary states [80,81], their capacity to imagine causes for different scenarios [62], and the presence of other psychiatric symptoms, such as problem gambling [82]. This issue is somewhat beyond the scope of the current project, but future research should investigate cognitive and affective influences on probability estimates, and in particular whether some individuals give different estimates for different categories of events, for example social vs. health-related events [70,71]. This is especially important given the relatively high number of socially-relevant negative events included in the Probability Bias Measure.

## Generalisability

Here, we have assessed the psychometric properties of this measure in samples of students. Our laboratory focuses on students, due to their elevated vulnerability to depression [83,84] and the benefits of addressing young people's mental health problems at the beginning of their careers. However, our measure's psychometric properties may vary in other populations. Furthermore, Sabanci University students may not be representative of other students in Turkey: they tend to be more politically left-liberal, higher socioeconomic status, and urban than students elsewhere in the country. These issues may explain why negative events relating to social situations tended to perform well in Study 1: events related to ill-health or disability may be more concerning for older participants; events relating to being a victim of crime, or financial difficulty, may be more relevant to participants with lower socioeconomic status. In future research, we will continue to assess the validity and factor structure of the Probability Bias Measure in more diverse and more representative samples.

In particular, the validity of this measure in clinical samples remains untested. Probability bias is most strongly associated with depression: in patients with severe depressive symptoms, and lower positive mood, the psychometric properties of our measure may vary. For example, probability bias might correlate more strongly with negative mood in such individuals, since their negative mood will tend to be elevated, and their positive mood relatively low, when compared to healthy individuals. In future studies, we will compare the scores of different groups of patients and nonpatients, to further probe the criterion validity of our measure. In particular, it remains to be tested whether the properties of this measure specifically, and probability bias in general, vary among depressive disorders, among anxiety disorders, and among bipolar disorders.

Similarly, we have tested the validity of our measure in Turkey; it remains to be seen whether the measure performs adequately in other contexts and cultures. Given that depression itself, both as a concept and as a disorder, can vary somewhat between cultures [85], this is not a trivial concern. We hope that future research will examine cross-cultural variation in the nature of probability bias.

Finally, we note that we have validated the measure using other self-report instruments; since all measures are self-report, this might exaggerate the correlations among them. It would also be instructive to validate the measure using behavioural measures of probability bias [86].

**Evidence for validity in English.** Since we conduct most of our research in Turkey, we have prepared and tested our measure in Turkish. However, the measure may perform adequately in English translation, and we present both Turkish and English versions of the measure in S1 File. Full Measure. Please note, however, that the English version of the Probability Bias Measure has not yet been formally validated.

Although we have not yet published research using this particular set of events in English, we have used the same general approach to measuring probability bias, and in particular the same response scale, in several recent projects [11,19,20]. In these seven studies, positive and negative events always showed good internal consistency, and the probability bias index correlated robustly with depressive and anxious symptom severity (although the relationship with depressive symptoms is more direct; see [11]). This was despite the fact that we used different sets of events in these projects, which once again supports our case that biases in expectancies can be quite general. However, these studies also employed samples of unselected, relatively-healthy students, and it remains to be seen whether these results would generalise to other populations.

**Self-report measure, or cognitive task?**

Throughout this article, we have referred to our probability bias *measure*. This is because our measure requires participants to quantify their judgements about probabilities and risks of specific events, rather than rate their agreement with general statements. Therefore, our measure straddles the line between self-report scale and behavioural, cognitive task.

This may give probability bias in general, and our measure in particular, the advantage of assessing the superordinate optimism/pessimism construct rather directly (see [87]). Other scales for measuring optimism/pessimism [e.g., 75], or future expectancies [88,89], ask participants to rate their agreement with good vs. bad statements about the future. Therefore, such scales may, to some extent, assess how much the participant perceives themselves to be optimistic/pessimistic, rather than how optimistic/pessimistic their snap judgements of probability are. This may partly explain why correlations between these scales and probability bias are sometimes weaker than one might expect [6]. Certainly, our measure complements existing scales of optimistic/pessimistic future thinking.

## Conclusion

Recent years have seen a notable, if belated, surge of interest in future thinking and its relevance to psychopathology [e.g., 15,16,90]. Research on probability bias began earlier than this, but has been characterised by ad hoc measures and inconsistent methods. We hope our measure proves useful to other researchers. We also hope that by paying attention to the psychometric properties and factorial structure of probability estimates, we can motivate more theoretical research on the cognitive and functional nature of probability bias, and of future thinking biases in general.

## Supporting information

**S1 File. Full measure.**
(DOCX)

**S2 File. Additional events.**
(DOCX)

## Acknowledgments

We are grateful to Nebi Sümer for his comments on the manuscript.

## Author contributions

**Conceptualization:** Robert W. Booth.

**Data curation:** Robert W. Booth.

**Formal analysis:** Robert W. Booth.

**Investigation:** Robert W. Booth, Gubse N. Aydın, Beril Başara, Ceren Yılmaz.

**Methodology:** Robert W. Booth.

**Project administration:** Robert W. Booth.

**Supervision:** Robert W. Booth.

**Writing – original draft:** Robert W. Booth.

**Writing – review & editing:** Gubse N. Aydın, Beril Başara, Ceren Yılmaz.

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
