## [Decision Letter · Decision Letter 0]

13 Jan 2026

Dear Dr. Booth,

Thank you for submitting your manuscript to PLOS ONE. After careful consideration, we feel that it has merit but does not fully meet PLOS ONE’s publication criteria as it currently stands. Therefore, we invite you to submit a revised version of the manuscript that addresses the points raised during the review process.

We look forward to receiving your revised manuscript.

Kind regards,

Kamalakar Surineni, MD, MPH

Guest Editor

PLOS One

Journal Requirements:

Reviewers' comments:

Reviewer's Responses to Questions

**Comments to the Author**

1. Is the manuscript technically sound, and do the data support the conclusions?

Reviewer #1: Yes

Reviewer #2: Yes

Reviewer #3: Yes

Reviewer #4: Yes

2. Has the statistical analysis been performed appropriately and rigorously?

Reviewer #1: Yes

Reviewer #2: Yes

Reviewer #3: I Don't Know

Reviewer #4: I Don't Know

3. Have the authors made all data underlying the findings in their manuscript fully available?

Reviewer #1: Yes

Reviewer #2: Yes

Reviewer #3: Yes

Reviewer #4: Yes

4. Is the manuscript presented in an intelligible fashion and written in standard English?

Reviewer #1: Yes

Reviewer #2: Yes

Reviewer #3: Yes

Reviewer #4: Yes

Reviewer #1: 1. When measuring human traits on a continuum, I think it’s relatively unlikely that there will be important differences by language or culture. You can note where you performed your study, but you can likely deemphasize the specific context. It’s a little distracting.

2. Twenty-four items is long. I understand that this may improve performance in research context. But for everyday use with patients a shorter version would be useful and almost certainly adequate. I suggest an unplanned analysis where you use the best performing 4 to 6 items in a shorter version and compare it to the longer version.

3. Line 160. You don’t have sufficient precision for 4 significant digits. Two significant digits is more appropriate. You can round the age to whole numbers. At line 253 and other areas there are numbers to adjust to likely 2, but certainly not more than 3 significant digits.

4. Study 2. Is it possible that the measures you are placing in a multivariable model are sufficiently related to cause distortion in the model. Have you considered alternative statistical techniques to account for anticipated collinearity? For instance, cluster or profile analyses?

Reviewer #2: Strengths:

Relevance and significance: The topic is clinically meaningful and addresses a recognized gap in current knowledge.

Clear study design: The research objectives are well defined, and the chosen methodology is appropriate for addressing them.

Logical structure: The manuscript follows a clear and coherent structure, making it easy to follow.

Interpretation of findings: Results are thoughtfully discussed and placed in appropriate context within the existing literature.

Ethical considerations: Ethical approval and consent procedures are clearly described and appropriate.

Major Comments:

1. Methodological Clarity

Additional detail regarding participant selection, data collection procedures, and/or analytic framework would strengthen transparency and reproducibility.

2. Context and Generalizability

The Discussion would benefit from a clearer explanation of how the findings may (or may not) be transferable to other clinical or geographic settings.

3. Integration With Existing Literature

While the literature review is adequate, deeper comparison with key prior studies would further strengthen the manuscript’s contribution.

Minor Comments:

1. Clarity and Style

Minor grammatical and stylistic edits would improve readability, particularly in the Discussion section.

2. Terminology

Ensure consistent use of key terms throughout the manuscript.

3. Tables/Figures

Consider adding or refining summary tables/figures to enhance clarity of results, if applicable.

Reviewer #3: This manuscript reports the development/validation of a brief Probability Bias Measure (12 positive + 12 negative future events) in Turkish-speaking university students, across two studies/samples. The authors report (a) mostly one-factor structures for positive and negative item sets via principal axis factoring on polychoric matrices, (b) good internal consistency, (c) good test–retest reliability (Study 2; ~4-week interval), and (d) convergent/discriminant validity via correlations with depressive/anxious symptoms, mood, hopelessness, optimism, and partial correlations showing probability bias explains variance in depression/positive mood beyond hopelessness and optimism. Overall, the project is useful and generally well-motivated. I have the following recommendations:

1. Scale structure

- The authors only used exploratory factor analysis.

- They did not test the structure using confirmatory methods.

- This makes it hard to say the scale truly has one clear factor.

Suggested change

- Add a confirmatory factor analysis if possible.

- If not, use more cautious language about the scale structure.

2. Item content

- Most negative items focus on social judgment or embarrassment.

- This may measure social anxiety rather than general probability bias.

Suggested change

- Acknowledge this limitation clearly.

- Consider adding items about other negative events (health, accidents, loss).

- If items stay the same, describe the scale as socially focused.

3. Test–retest timing

- Data were collected over a long time period.

- Mood and stress may have changed during this time.

- Dropout between time points is not fully described.

Suggested change

- Report how many people dropped out.

- Compare people who completed both surveys with those who did not.

4. Additional limitations to be mentioned/addressed

- The main score is a difference score, which can be hard to interpret.

- Only university students were studied; results may not apply to clinical groups.

- Excluding people who failed attention checks may bias the sample.

- Translation and cultural adaptation steps are not fully explained.

- All data are self-report, which can inflate correlations.

5. Additional suggestions:

- Clearly state how missing data were handled.

- Clarify whether questionnaire responses were treated as ordinal or continuous.

- Use cautious terms like “preliminary validation” instead of “validated.”

6. References:

- Replace the citation listed as “manuscript under review” with a publicly accessible preprint or remove it if no public version exists.

- Correct the journal name “Clinial Psychology Review” to Clinical Psychology Review.

- Correct the publisher name “Guildford Press” to Guilford Press.

- Remove duplicate “doi:” labels so each reference contains only one DOI.

- Standardize all DOIs to the correct format: https://doi.org/10.xxxx.

- Fix malformed DOI links that are missing characters (e.g., “https//doi.org”).

- Complete any references missing page numbers, issue numbers, or DOIs where available.

- Clearly label any preprints as preprints and avoid presenting them as peer-reviewed evidence.

- Review the entire reference list for consistent formatting according to PLOS ONE guidelines.

Reviewer #4: Line 416- Error variance may influence results

Line 448- While the measure was tested in Turkish, its performance in English translation has not been thoroughly validated

Line 441- Applicability in specific types of anxiety or depression

Line 460- Results may not apply to individuals with severe psychopathology or to non-student populations

.

Reviewer #1: **Yes:** David RingDavid RingDavid RingDavid Ring

Reviewer #2: **Yes:** VENKATA VIJAYA K DALAIVENKATA VIJAYA K DALAIVENKATA VIJAYA K DALAIVENKATA VIJAYA K DALAI

Reviewer #3: **Yes:** Nikhil TondehalNikhil TondehalNikhil TondehalNikhil Tondehal

Reviewer #4: **Yes:** Anoop NarahariAnoop NarahariAnoop NarahariAnoop Narahari

---

## [Author Response · Author response to Decision Letter 1]

10 Feb 2026

Below, we outline our revisions and responses to the reviewers’ comments. Line numbers refer to the tracked-changes version of our manuscript.

Reviewer 1

1. When measuring human traits on a continuum, I think it’s relatively unlikely that there will be important differences by language or culture. You can note where you performed your study, but you can likely deemphasize the specific context. It’s a little distracting.

We agree, but linguistic and cultural differences remain to be tested, and we did not wish to overstate the generalisability and validity of our measure. We have clarified the relevance of the Turkish context in our Introduction – it is where we conduct most of our research – and deemphasised it where we could. In particular, note that we have amended the title, in order to emphasise the construct we are attempting to measure, more than the context in which we are measuring it.

2. Twenty-four items is long. I understand that this may improve performance in research context. But for everyday use with patients a shorter version would be useful and almost certainly adequate. I suggest an unplanned analysis where you use the best performing 4 to 6 items in a shorter version and compare it to the longer version.

This is a great point. Certainly, we hope our measure can eventually be useful for clinicians. However, at this stage we feel that a practice-oriented short version would be premature. As discussed below and in our revised General Discussion, the validity of our measure in clinical samples remains to be tested, and in particular, the measure may be biased towards socially-relevant events. We are currently collecting data from clinical patients with our measure, and we will continue to examine the factorial structure of probability estimates in clinical and healthy populations. Once we have completed this work, we will develop a robust and reliable short pen-and-paper form of the Probability Bias Measure for applied research and practice.

3. Line 160. You don’t have sufficient precision for 4 significant digits. Two significant digits is more appropriate. You can round the age to whole numbers. At line 253 and other areas there are numbers to adjust to likely 2, but certainly not more than 3 significant digits.

We now present all ages as whole numbers, and all scale total scores with one decimal place.

4. Study 2. Is it possible that the measures you are placing in a multivariable model are sufficiently related to cause distortion in the model. Have you considered alternative statistical techniques to account for anticipated collinearity? For instance, cluster or profile analyses?

We presume the reviewer is referring to the partial correlations. In assessing this, we realised that we had inadvertently calculated these correlations based on the combined dataset, not the complete Part 1 dataset. This has now been corrected, and our conclusions are unchanged.

When we repeated these analyses in conventional regression models, we saw no evidence of multicollinearity (all tolerances > .49). We have now mentioned this in our revised manuscript (lines 419-420).

Reviewer 2

Additional detail regarding participant selection, data collection procedures, and/or analytic framework would strengthen transparency and reproducibility.

We have ensured that we have reported all these details in as much detail as possible.

The Discussion would benefit from a clearer explanation of how the findings may (or may not) be transferable to other clinical or geographic settings.

This is an important point, and we have expanded this section of the General Discussion to more thoroughly discuss these issues (lines 485-499).

While the literature review is adequate, deeper comparison with key prior studies would further strengthen the manuscript’s contribution.

We now compare the psychometric properties of our measure with those of other similar measures in the literature (lines 430-431), and we also suggest – with citations – potential moderators of our measure’s correlations with symptom severity and mood (lines 462-464).

Minor grammatical and stylistic edits would improve readability, particularly in the Discussion section.

We have paid careful attention to the readability of our revised manuscript, and we believe it is improved.

Ensure consistent use of key terms throughout the manuscript.

We have paid careful attention to this issue. The terminology is inconsistent in the literature, so we have strived to clarify exactly what we mean by each term.

Reviewer 3

1. Scale structure

- The authors only used exploratory factor analysis.

- They did not test the structure using confirmatory methods.

- This makes it hard to say the scale truly has one clear factor.

Suggested change

- Add a confirmatory factor analysis if possible.

- If not, use more cautious language about the scale structure.

The reviewer is quite correct here. As we have stated in the manuscript (lines 148-153), the factor structures of our events were not the primary focus of these studies (our samples were rather small for assessing this), and were also somewhat unclear. So, we felt that confirmatory analyses would be unhelpful. As the reviewer suggests, we have now carefully moderated our language when we discuss the factor structures of our events.

2. Item content

- Most negative items focus on social judgment or embarrassment.

- This may measure social anxiety rather than general probability bias.

Suggested change

- Acknowledge this limitation clearly.

- Consider adding items about other negative events (health, accidents, loss).

- If items stay the same, describe the scale as socially focused.

This is an important point, and we did not emphasise it as much as we intended in our original manuscript. In our revised manuscript, we specifically call attention to this issue in the Results and Discussions of Study 1 (lines 258-259; 283-284) and Study 2 (lines 448-449), and in the General Discussion (lines 468-470; 479-484).

3. Test–retest timing

- Data were collected over a long time period.

- Mood and stress may have changed during this time.

- Dropout between time points is not fully described.

Suggested change

- Report how many people dropped out.

- Compare people who completed both surveys with those who did not.

We now explicitly state how many participants dropped out in the Participants section for Study 2. We also state that there were no significant differences between returning participants and dropouts on any of the demographic or Time-1 study variables we examined (lines 313-315).

4. Additional limitations to be mentioned/addressed

- The main score is a difference score, which can be hard to interpret.

In the literature, the bias index – the difference score – actually tends to show more consistent correlations with other variables than do the probability estimates themselves. This may be due to the superior reliability of difference scores between negatively correlated variables. We mention this point in lines 117-122, and reiterate it for clarity in lines 270-272 and 428-429.

- Only university students were studied; results may not apply to clinical groups.

This is an important point, and was also mentioned by other reviewers. We have discussed this issue in our revised General Discussion (lines 485-494).

- Excluding people who failed attention checks may bias the sample.

This is an interesting point. The probability estimation task in Study 1 was rather long, so a substantial number of participants failed the attention check at the end of it. While excluding them may bias the sample, including their responses – which are known to be unreliable – is very likely to weaken, and perhaps also to bias, the results.

Given the small numbers of participants who failed the attention checks in Study 2, we feel the risk is negligible in this case.

- Translation and cultural adaptation steps are not fully explained.

This process was handled by the native Turkish-speaking authors. Some events were in Turkish already, having been prepared for previous projects; others were checked and adapted during the process of forward-translation. All authors approved the final versions of each event. We now clarify these details in lines 178-181.

- All data are self-report, which can inflate correlations.

This is difficult to avoid with this type of measure. We now acknowledge the issue in lines 500-503.

5. Additional suggestions:

- Clearly state how missing data were handled.

We apologise for omitting this important information. It is now included in lines 222-227.

- Clarify whether questionnaire responses were treated as ordinal or continuous.

This is clarified in line 230.

- Use cautious terms like “preliminary validation” instead of “validated.”

The reviewer is quite correct here, and we have moderated our language throughout, especially in the Abstract and General Discussion (lines 455-457).

6. References:

- Replace the citation listed as “manuscript under review” with a publicly accessible preprint or remove it if no public version exists.

- Correct the journal name “Clinial Psychology Review” to Clinical Psychology Review.

- Correct the publisher name “Guildford Press” to Guilford Press.

- Remove duplicate “doi:” labels so each reference contains only one DOI.

- Standardize all DOIs to the correct format: https://doi.org/10.xxxx.

- Fix malformed DOI links that are missing characters (e.g., “https//doi.org”).

- Complete any references missing page numbers, issue numbers, or DOIs where available.

- Clearly label any preprints as preprints and avoid presenting them as peer-reviewed evidence.

- Review the entire reference list for consistent formatting according to PLOS ONE guidelines.

We thank the reviewer for their careful attention to these. We have reviewed our Reference list, and made sure that all are correct.

Reviewer 4

Line 416- Error variance may influence results

The reviewer is, of course, quite right here: if the overall error variance in probability estimates was higher in Study 1 than it was in Study 2, this should have had an overall weakening effect, and decreased the correlations between the probability bias index and the other validation scales. This was not the case, so we have removed this statement from our revised Discussion of Study 2 (line 442).

Line 448- While the measure was tested in Turkish, its performance in English translation has not been thoroughly validated

It has not, and we have updated this section of our revised General Discussion, to make this clearer (lines 508-509).

Line 441- Applicability in specific types of anxiety or depression

This is an excellent point, and we have added it to our expanded discussion of the performance of our measure in clinical samples (lines 490-494).

Line 460- Results may not apply to individuals with severe psychopathology or to non-student populations

Again, we have now included an expanded discussion of these issues in our revised General Discussion (lines 470-494).

In summary, we have done our utmost to accommodate all the reviewers’ feedback, and we believe our revised manuscript is much stronger as a result. We thank you and the reviewers for your diligence and helpful commentaries, and we hope our revised manuscript now meets your criteria for publication in PLOS ONE.

---

## [Decision Letter · Decision Letter 1]

25 Mar 2026

Psychometric properties of the Probability Bias Measure

PONE-D-25-54306R1

Dear Dr. Booth,

We’re pleased to inform you that your manuscript has been judged scientifically suitable for publication and will be formally accepted for publication once it meets all outstanding technical requirements.

Kind regards,

Kamalakar Surineni, MD, MPH

Guest Editor

PLOS One

Additional Editor Comments (optional):

Reviewers' comments:

Reviewer's Responses to Questions

**Comments to the Author**

Reviewer #1: All comments have been addressed

Reviewer #2: All comments have been addressed

Reviewer #4: All comments have been addressed

2. Is the manuscript technically sound, and do the data support the conclusions?

Reviewer #1: Yes

Reviewer #2: Yes

Reviewer #4: Yes

3. Has the statistical analysis been performed appropriately and rigorously?

Reviewer #1: Yes

Reviewer #2: Yes

Reviewer #4: Yes

4. Have the authors made all data underlying the findings in their manuscript fully available?

Reviewer #1: Yes

Reviewer #2: Yes

Reviewer #4: Yes

5. Is the manuscript presented in an intelligible fashion and written in standard English?

Reviewer #1: Yes

Reviewer #2: Yes

Reviewer #4: Yes

Reviewer #1: (No Response)

Reviewer #2: The manuscript presents a well-conducted preliminary psychometric evaluation of the Probability Bias Measure in Turkish-speaking student samples. The statistical analyses are generally appropriate and clearly reported, including the use of polychoric correlations for ordinal data, internal consistency estimates, test–retest reliability, and convergent and discriminant validity analyses. The findings support the authors’ conclusions within the limits of a non-clinical student population. The manuscript is clearly written, logically structured, and presented in standard academic English.

The authors have been appropriately cautious in framing their conclusions as preliminary and population-specific. The acknowledgment of limitations—particularly the reliance on student samples, the socially focused content of many negative items, the absence of confirmatory factor analysis, and the need for validation in clinical and cross-cultural contexts—is appreciated. The data availability statement is transparent, and the provision of anonymized datasets enhances reproducibility. Ethical approval and consent procedures are adequately described.

Suggestions for strengthening the manuscript include: (1) clarifying the rationale for relying solely on exploratory factor analysis and outlining plans for future confirmatory work; (2) further discussing how socially weighted negative items may influence associations with social anxiety versus broader depressive constructs; and (3) expanding on potential clinical applications and thresholds for interpretation in applied settings.

No concerns regarding dual publication or research ethics are evident. Overall, this is a thoughtful and methodologically sound contribution that advances research on expectancy-related cognitive biases.

Reviewer #4: Line 416 - Error variance may influence results: The authors removed the statement about error variance

Performance in English translation: Authors acknowledged that the English version of the measure has not been thoroughly validated

Applicability in specific types of anxiety or depression: The authors expanded the discussion on this point

Results may not apply to individuals with severe psychopathology or non-student populations: authors acknowledged this limitation and expanded discussion.

.

Reviewer #1: **Yes:** David Ring MDDavid Ring MDDavid Ring MDDavid Ring MD

Reviewer #2: **Yes:** VENKATA VIJAYA K DALAIVENKATA VIJAYA K DALAIVENKATA VIJAYA K DALAIVENKATA VIJAYA K DALAI

Reviewer #4: **Yes:** Anoop NarahariAnoop NarahariAnoop NarahariAnoop Narahari

---

## [Editor Report · Acceptance letter]

PONE-D-25-54306R1

PLOS One

Dear Dr. Booth,

I'm pleased to inform you that your manuscript has been deemed suitable for publication in PLOS One. Congratulations! Your manuscript is now being handed over to our production team.

Kind regards,

on behalf of

Dr. Kamalakar Surineni

Guest Editor

PLOS One